

# Climate-groundwater dynamics inferred from GRACE and the role of hydraulic memory

**Simon Opie[1,*], Richard G. Taylor[1], Chris M. Brierley[1], Mohammad Shamsudduha[2] and**

**Mark O. Cuthbert[3,4]**

[1] Department of Geography, University College London, London, UK

[2] Department of Geography, University of Sussex, Falmer, Brighton, UK

[3] School of Earth and Ocean Sciences, Cardiff University, Cardiff, UK

[4] Connected Waters Initiative Research Centre, University of New South Wales, Sydney, New

South Wales, Australia

[*]  Corresponding author: Simon Opie (simon.opie.18@ucl.ac.uk)

**Abstract**

Groundwater is the largest store of freshwater on Earth after the cryosphere and provides a

substantial proportion of the water used for domestic, irrigation and industrial purposes. Knowledge

of this essential resource remains incomplete, in part, because of observational challenges of scale

and accessibility. Here we examine a 14-year period (2002-2016) of GRACE observations to

investigate climate-groundwater dynamics of 14 tropical and sub-tropical aquifers selected from

WHYMAP's 37 large aquifer systems of the world. GRACE-derived changes in groundwater storage

resolved using GRACE JPL Mascons and the CLM Land Surface Model are related to precipitation

time series and regional-scale hydrogeology. We show that aquifers in dryland environments exhibit

long-term hydraulic memory through a strong correlation between groundwater storage changes

and annual precipitation anomalies integrated over the time series; aquifers in humid environments

show short-term memory through strong correlation with monthly precipitation. This classification is

consistent with estimates of Groundwater Response Times calculated from the hydrogeological

properties of each system, with long (short) hydraulic memory associated with slow (rapid) response

times. The results suggest that groundwater systems in dryland environments may be less sensitive

to seasonal climate variability but vulnerable to long-term trends from which they will be slow to

recover. In contrast, aquifers in humid regions may be more sensitive to seasonal climate

disturbances such as ENSO-related drought but may also be relatively quick to recover. Exceptions to

this general pattern are traced to human interventions through groundwater abstraction. Hydraulic

memory is an important factor in the management of groundwater resources, particularly under

climate change.



### 1.0. Introduction:

The availability of freshwater is essential for sustaining human life, economic security, and access to the benefits of a wide range of ecosystem services (Taylor et al., 2013a). After the cryosphere, groundwater is the second largest store of freshwater on the planet supplying 36% of domestic water, 42% of irrigation for agriculture and 27% of industrial water use (Döll et al., 2012). Baseflow from groundwater sustains rivers and wetlands in the absence of rainfall and is therefore fundamentally important to the ecology of semi-arid and arid regions in particular (Alley et al., 2002; Graaf et al., 2019). Future climate change in which anthropogenic emissions of greenhouse gases transform patterns of natural variability, together with substantial socio-economic change, predicate that management of freshwater resources will become a critical task (Famiglietti, 2014). In a climate where it is broadly predicted that 'wet gets wetter, dry gets drier' (Trenberth, 2011), water storage at and below the land surface will be a vital tool in enabling successful adaptation to the changing global environment (Damkjaer and Taylor, 2017; Wada, 2016).

Despite the importance of groundwater there are considerable gaps in current knowledge and understanding (Güntner et al., 2007). Direct observations of groundwater are sparse in relation to its geographical scale so most global or regional groundwater data are based on output from large-scale models. These include global hydrological models (GHMs) (Sood and Smakhtin, 2015) or land- surface models (LSMs) (Bierkens, 2015; Overgaard et al., 2006; Wood et al., 2011) for which there are often insufficient data available to constrain or calibrate (Döll et al., 2016). Model simulation of key processes such as soil hydrodynamics and groundwater recharge is therefore based on theoretical frameworks rather than field data (Scanlon et al., 2002). As a result, there is also considerable uncertainty about climate-groundwater dynamics. Recent work in this area has either focused on localised observations of changes in Groundwater Storage (ΔGWS) from piezometry (Cuthbert et al., 2019b) or occurred adjacent to large centres of population where human intervention, through extraction of groundwater by pumping, can greatly influence observational measurements (Scanlon et al., 2018). In the context of a recent, rapid escalation in groundwater abstraction of ~15% per decade from 1960 to 2010 (Wada et al., 2014), an understanding of climate-groundwater dynamics, supported by large-scale observational data, is required to inform sustainable access to groundwater resources (Taylor et al., 2009).

In response to the lack of in situ field observations, remote-sensing by satellite is increasingly being utilised to expand the scope of observational data available to Earth sciences (Acker and Leptoukh, 2007). An important advance in the quality of global data for hydrological studies has come from the Gravity Recovery and Climate Experiment (GRACE), a collaboration





between the National Aeronautics and Space Administration (NASA) in the USA and the German

Aerospace Centre (DLR) launched in March 2002 (Tapley et al., 2004). Completed sets of ~monthly

measurements are used to derive the changes in mass at the Earth's surface and from these data

mass fluxes can be extracted that directly relate to the hydrosphere. Over land, the flux is expressed

as a change in Total Water Storage (ΔTWS) at a spatial resolution of ~300km and with an expected

accuracy of better than 2 cm equivalent water height (EWH) (Tapley et al., 2004). GRACE ceased

operation due to battery failure in mid-2016 having created a record of 163 monthly gravity

solutions (Tapley et al., 2019). Although GRACE operated for ten years longer than anticipated at its

launch, it is a relatively brief dataset in relation to large-scale climate patterns impacting the global

hydrological system with frequencies of several years or decades (e.g. Pacific Decadal Oscillation

(PDO), Atlantic Multidecadal Oscillation (AMO)). Nevertheless, inter-annual periodicities associated

with the El Niño Southern Oscillation (ENSO) and the Antarctic Circumpolar Wave (ACW) have been

detected (Mémin et al., 2015; Ni et al., 2018; Phillips et al., 2012).

Intrinsic parameters of GRACE data effectively define the spatial and temporal dimensions of

this study but there are additional constraints related to the derivation of ΔGWS data from GRACE

ΔTWS that also need to be considered. The sub-division of GRACE ΔTWS into its component parts,

including ΔGWS, requires the application of GHM or LSM output that is itself subject to associated

uncertainty, as already noted (Döll et al., 2014). It has been demonstrated that there is relatively

poor correlation between GRACE and GHMs/LSMs in the evaluation of ΔTWS, with significant

discrepancies at the basic level of whether storage trends are increasing or decreasing (Scanlon et

al., 2018). These findings have been confirmed with reference to regional piezometric groundwater

measurements from tropical aquifers in Africa (Bonsor et al., 2018). Thus, the application of GRACE

data to ΔGWS implies three distinct areas of uncertainty: in the processing of the GRACE signal,

accuracy of GHM/LSM model projections and mutual consistency of the observed (GRACE) and

modelled (GHM/LSM) data (Long et al., 2015).

This study investigates the spatio-temporal properties of climate-groundwater dynamics

using a subset of the 37 Large Aquifer Systems of the World (LASW) as defined by the Worldwide

Hydrogeological Mapping and Assessment Programme (WHYMAP) ("BGR - WHYMAP - Large

Aquifers," 2008), and shown in **Figure S1**. This subset comprises aquifers that lie broadly within the

tropics and sub-tropics climate variability is mostly defined by rainfall (Shepherd, 2014). The 14

aquifers selected are listed in **Table 1** together with their key characteristics including Aridity Index

(AI) calculated from the Consultative Group for International Agriculture Research's Consortium for

Spatial Information (CGIAR-CSI) Global-Aridity Dataset (Trabucco and Zomer, 2019), shown in **Figure

1**. Following the work of (Shamsudduha and Taylor, 2019), the groundwater storage response to



regional climate variability for these 14 large scale aquifer systems is investigated using ΔGWS data extracted from the whole of the available GRACE ΔTWS time series (August 2002 – July 2016)

together with climate data that are defined by the areal extent of each of the aquifer systems.

Several studies have used GRACE data to examine storage changes within a particular GW system e.g. (Becker et al., 2010; Bonsor et al., 2018; Chen et al., 2016, 2010; Henry et al., 2011; Z. Huang et al., 2015; Ramillien et al., 2014; Shamsudduha et al., 2017, 2012; Tiwari et al., 2009; Xavier et al., 2010; Yeh et al., 2006). Here, we examine the dynamics of climate-groundwater interactions

inferred from the underlying patterns of large-scale ΔGWS in response to extremes of precipitation. We find that hydraulic memory (HM) is a key component in the classification of groundwater responses to climate variability. We then seek to reconcile the results with reference to the physical characteristics of individual aquifer systems (Cuthbert et al., 2019a) whilst accounting for anomalous responses in ΔGWS to climate variability.


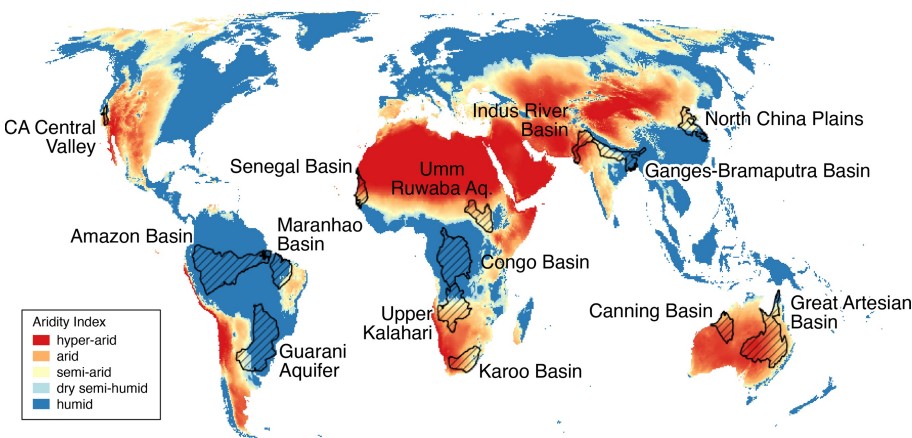

**Figure 1: 14 of the World's Large-scale Aquifers** (the Study Aquifers) overlaid on CGIAR-CSI Global-Aridity
dataset *(Trabucco & Zomer, 2019).*



| WHYMAP Aquifer Number | Aquifer name | Continent | Population (millions) | Aquifer area (km²) | Proportion of Irrigation GW fed (%) | Climate zone based on aridity index | Mean (2002-16) annual precipitation (mm) | Rainfall variability (%) |
|---|---|---|---|---|---|---|---|---|
| 5 | Senegal-Mauritanian Basin | Africa | 17.77 | 295k | 1.0 | Semi-Arid | 540 | 14.6 |
| 8 | Umm Ruwaba Aquifer | Africa | 10.52 | 509k | 0.0 | Semi-Arid | 789 | 10.7 |
| 10 | Congo Basin | Africa | 34.74 | 1.49m | 0.0 | Humid | 1566 | 5.6 |
| 11 | Upper Kalahari-Cavelai-Zambezi Basin | Africa | 6.02 | 1.00m | 0.1 | Semi-Arid | 819 | 10.0 |
| 13 | Karoo Basin | Africa | 14.53 | 568k | 2.1 | Semi-Arid | 479 | 17.6 |
| 16 | California Central Valley Aquifer System | North America | 8.10 | 71k | 57.8 | Semi-Arid | 515 | 32.0 |
| 19 | Amazon Basin | South America | 8.93 | 2.28m | 1.0 | Humid | 2505 | 8.3 |
| 20 | Maranhao Basin | South America | 10.81 | 593k | 32.6 | Humid | 1502 | 15.7 |
| 21 | Guarani Aquifer (Parana Basin) | South America | 47.84 | 1.83m | 20.5 | Humid | 1450 | 10.6 |
| 23 | Indus River Basin | Asia | 155.85 | 308k | 31.0 | Arid | 375 | 16.2 |
| 24 | Ganges-Brahmaputra Basin | Asia | 596.44 | 616k | 55.8 | Humid | 1391 | 12.1 |
| 29 | North China Plains Aquifer System | Asia | 336.70 | 439k | 37.1 | Dry Sub-Humid | 826 | 10.0 |
| 36 | Great Artesian Basin | Australia | 0.20 | 1.77m | 0.9 | Arid | 444 | 28.9 |
| 37 | Canning Basin | Australia | 0.01 | 433k | 0.4 | Arid | 443 | 21.2 |


**Table 1:** Characteristics of the 14 Aquifer Systems selected for the study according to the WHYMAP and CGIAR-CSI databases with statistics giving (L to R): total number of resident population, aquifer area, proportion of irrigation GW-fed, mean aridity index classification (Trabucco and Zomer, 2019), mean annual rainfall and mean variability in annual rainfall.






*Methods:*


### 2.1. *GWS derived from GRACE data:*

Mass fluxes relating to the hydrosphere contained in the GRACE land-signal measurement of
changes in the Earth's gravitational field are defined as ΔTWS. In order to obtain information relating
specifically to groundwater, this signal is separated into the component parts that comprise TWS,
generally represented as:


$$\Delta TWS = \Delta GWS + \Delta SWS + \Delta SMS + \Delta SNS \qquad\qquad (1)$$

where SWS is surface water storage, SMS is soil moisture storage and SNS is snow-water equivalent
storage. ΔGWS is then derived from ΔTWS according to the following equation:


$$\Delta GWS = \Delta TWS - (\Delta SWS + \Delta SMS + \Delta SNS) \qquad\qquad (2)$$

The locations of the 14 aquifers are outside areas where changes in snow-water equivalent
substantially impact ΔTWS (Getirana et al., 2017).  ΔSNS can consequently be omitted so that Eq. (2)
can be rewritten for the purposes of this particular study as:

$$\Delta GWS = \Delta TWS - (\Delta SWS + \Delta SMS) \qquad\qquad (3)$$

Since GRACE started transmitting, several solutions have been developed for analysing and
producing GRACE ΔTWS data to increasing levels of accuracy, with the intention that the data be
readily and freely available for research (Landerer and Swenson, 2012). In this instance, three
different products were drawn from Shamsdduha and Taylor (2019), two of which are spherical
harmonics (SH) solutions comprising CSR Land (version RL05.DSTvSCS1409) from the Jet Propulsion
Laboratory (JPL) at NASA and CNES/GRGS (version RL03-v1) from the French Centre National
d'Etudes Spatiales, and one JPL-Mascon (version RL05M 1.MSCNv01) from JPL-NASA. To derive
ΔTWS, all GRACE solutions require additional processing that include corrections for glacial isostatic
rebound and atmospheric mass variation (Landerer and Swenson, 2012). SH solutions also require
spatial filtering (or 'de-striping') whereas JPL-Mascon does not as it directly converts the GRACE


signal into mass concentration blocks (Mascons), rendering monthly gravitational fields directly as 3º×3º gridded spatial components to reduce errors (Watkins et al., 2015).

On inspection, the divergence between the 3 ΔTWS datasets was significant when summed over the time series. The relatively large coefficient of variance (COV), -104%, calls into question use of an ensemble mean for this study. Such an approach may be appropriate for the use of SH

products alone (Sakumura et al., 2014) but it is preferable not to combine SH products and Mascons (Landerer, pers. comm.). Consequently we rely solely on the JPL-Mascon dataset possessing a better signal-to-noise ratio and potentially less error (Scanlon et al., 2016; Watkins et al., 2015; Xie et al., 2018). The employed JPL-Mascon dataset has been spatially sampled at a 0.5º grid using dimensionless scaling factors provided as 0.5º×0.5º bins derived from the CLM4.0 LSM (Long et al.,

2015; Wiese et al., 2016). GRACE ΔTWS is not a time-invariant measure (Wahr et al., 1998) and in the standard datasets all anomalies are given with respect to a baseline which is the mean over the period January 2004 to December 2009 (JPL NASA, 2019). Here, the completed available GRACE ΔTWS time series is examined with respect to climate anomalies in the same timeframe. Consequently, the JPL-Mascon ΔTWS dataset has been rescaled with respect to a time-mean taken

over the whole period of GRACE operation (08.2002 – 07.2016), which is the study reference period (SRP) (JPL NASA, 2019).

As set out in Eq. (3), datasets for ΔSMS and ΔSWS derived from LSMs are required to determine ΔGWS from ΔTWS since observational data at the spatio-temporal scales of this study do not exist. Datasets for the 14 aquifer systems were drawn from NASA's Global Land Data

Assimilation System (GLDAS) (Rodell et al., 2004) comprising the output from four different LSMs (Shamsudduha and Taylor, 2019): the Common [Community] Land Model (CLM, version 2.0), Noah (version 2.7.1), the Variable Infiltration Capacity (VIC) model (version 1.0), and Mosaic (version 1.0) (Rui and Beaudoing, 2019). As with ΔTWS, analysis of the four LSM datasets for ΔSWS+ΔSMS shows that their divergence summed over the entire time series is substantial, with a COV of 258%,

suggesting that a LSM-ensemble mean approach may also not be appropriate for this analysis. Further, the inter- and intra- model variability of ΔSWS in the LSM datasets , assessed as surface runoff (e.g. Shamsudduha and Taylor, 2019; Thomas et al., 2017), is much less significant than that of ΔSMS (inter-model COV 378%). In the absence of consideration of ΔSWS, groundwater recharge is primarily determined by the effect of evapotranspiration on moisture in the soil zone (Long and

Mahler, 2013). Therefore, for this study, modelling of ΔSMS is a key determinant of the outcomes for ΔGWS computed using Eq. (3) (de Vries and Simmers, 2002). Modelled soil profiles vary substantially in each of the 4 LSMs ranging in depth from 3.5m (Mosaic) to 1.9m (VIC) and, in vertical layers, from 10 (CLM) to 3 (VIC & Mosaic) (Rodell et al., 2004). CLM 2.0 (Bonan et al., 2002; Dai et al.,



2003) with 3.4m depth and 10 vertical layers features the most well developed soil model (Scanlon

et al., 2018), has been shown to perform well in comparative testing (Scanlon et al., 2018;

Spennemann et al., 2014). In addition, CLM has demonstrated appropriate variability in initial

ensemble model runs undertaken here, meaning that ΔSMS is almost always less than the

magnitude of ΔTWS thereby ensuring that ΔGWS estimates derived from Eq. (3) are not arbitrarily

high or low (Shamsudduha and Taylor, 2019). Therefore, this study employs a single model, CLM, for

ΔSMS and ΔSWS rather than adopting a LSM ensemble mean approach.

   **2.2.** *Climatology:*

Individual aquifer system shapefiles from the WHYMAP LASW were prepared as ASCII files and

uploaded to KNMI Climate Explorer (KNMI Climate Explorer, 2018). This allowed a range of climate

data to be extracted for the precise spatial boundaries of each system. In particular, precipitation

(PCP) data from the CRU TS4.03 dataset at 0.5º resolution (Climate Research Unit, University of East

Anglia, 2019) was obtained together with anomalies (PCPA) normalised for the SRP (2002-16). The

CRU TS4.03 datasets together with the ΔGWS derived from JPL-Mascon ΔTWS and CLM 2.0 ΔSMS &

ΔSWS, in accordance with Eq. (3), were used to create time series analyses to explore correlations

over different time and volume components through integration. In this respect the use of 'annual'

in this study implies the appropriate hydrological year.

          In order to calibrate the time series for each aquifer system prior to further analysis, the lag

between monthly PCP, as the primary climate-groundwater index, and monthly GRACE ΔTWS was

set by maximising the Pearson Correlation Coefficient (PCC) between the two datasets, validated by

point-wise verification of alignment of the time series. In the majority of cases, this comparison

showed ΔTWS lagging PCP by two months. The lag for the PCPA time series were set in the same

way with relation to ΔGWS but with the already determined PCP time series lag set as a minimum. In

the case of all aquifer systems except for the Congo, Canning and Indus River Basins, this procedure

resulted in a consistent lag being applied to all of the time series investigations of each aquifer.

Initial investigations also established that only relatively weak first-order correlations exist between

ΔTWS and other monthly observational climate data such as the self-calibrating Palmer Drought

Severity Index (PDSI-sc) (Wells et al., 2004) and Mean Temperature anomalies (CPC GHCN/CAMS

t2m analysis) (Fan and van den Dool, 2008). By comparison with both these measures, it appeared

that PCPA carried a stronger climate variability signal due to the tropical/sub-tropical location of the

selected aquifers (Allan et al., 2010; Shepherd, 2014). An analysis was then conducted to test for

correlations between ΔGWS and a series of measures of precipitation. Three separate time series of





precipitation were developed to examine the temporal response of the study LASW with respect to the process by which precipitation at the land surface contributes to ΔGWS:


1. PCP = monthly precipitation
2. PCPA = monthly precipitation anomalies with respect to the time-mean baseline for the study reference period (SRP – 2002-16)
3. ∫PCPA = cumulative monthly rainfall anomalies derived by integrating the PCPA time series


These monthly series were also summed to provide annual time series for each aquifer system. Correlation was measured using the PCC with statistical significance determined by a t-test with **α**=0.05 (Spearman, 1904). In addition, as previously stated, the CGIAR-CSI Global-Aridity dataset (Trabucco and Zomer, 2019) was obtained and a numerical AI for each aquifer was extracted as a

spatial mean value using QGIS. AI was used to place each aquifer into the climate zone classification specified by the dataset as set out in Table 1. Of the climate zones relating to the 14 aquifer systems, 3 are Arid, 5 are Semi-Arid, 1 is Dry Semi-Humid, giving 9 in total in dryland zones (Corvalán et al., 2005), and 5 are Humid.

**_2.3. Hydraulic Memory (HM):_**

In using cumulative rainfall anomalies, this study invokes the concept of system memory (Weber and Stewart, 2004). Several studies have considered the question of hydraulic or hydrologic memory, both as it impacts soil moisture including land/atmosphere dynamics (Castro et al., 2009; Lo and

Famiglietti, 2010; Wu et al., 2002), and groundwater (Currell et al., 2016; Cuthbert et al., 2019a; Güntner et al., 2007; Rodell and Famiglietti, 2001). Central to the definition of this 'memory' is that it represents the time taken for a system to re-equilibrate following a change in boundary conditions (Downing et al., 1974). In the case of an aquifer system, approximated to a one-dimensional flow of uniform diffusivity, the groundwater response time (GRT) is given by Eq. (4):


GRT = $L^2 S/\beta T$                                                (4)

where $L$ is a measure of the scale of the system, $S$ is the storativity, $\beta$ is a dimensionless constant and $T$ is transmissivity. Qualitatively Eq. (4) implies that long response times are characterised by large-

scale systems and/or low hydraulic diffusivity (i.e. combination of high $S$ and low $T$) (e.g. Kooi and Groen, 2003). An alternative approach to quantifying memory may be needed in more complex -



and realistic – multidimensional flow situations (see Cuthbert et al., 2019a). Nevertheless, Eq. (4) still provides a useful order of magnitude approximation. Here, it is helpful to consider the response time as a delay between system input and system output whereby the output state *H*, at time *t*, is
given by:

$$H(t) = \int_{-\infty}^{t} p(\tau)\, \theta(t - \tau)\, d\tau \qquad\qquad (5)$$

where $p(\tau)$ is the input state or function at time $\tau$, $(t-\tau)$ is the delay between output and input, and
$\theta$ is an Impulse Response Function (IRF), also known as a transfer function (Long and Mahler, 2013). The IRF is a multi-parameter function that is intended to model the properties of the system so that the output of the IRF determines the time, *t*, at which the state *H* is reached. The hydraulic memory is quantified by the length of time that the effect of the input persists in the system. As the IRF is commonly exponential, making the equilibrium state asymptotic, system memory can be defined as
the time interval at which the IRF is 95% complete. This approach has been successfully applied to modelling aquifer responses to precipitation validated by piezometry in both the USA (Long and Mahler, 2013) and the Netherlands (von Asmuth and Knotters, 2004). Alternatively, system memory may be defined as the length of time taken for the effect of the anomalous input to decay to 1/e of its starting value where this can be explicitly measured (Cuthbert et al., 2019a; Lo and Famiglietti,
2010). In relation to Eq. (5), chosen precipitation measures are $p(\tau)$ input functions, and ΔGWS represents *H(t)*, the output measure. The timestep, $\tau$, for each of the precipitation time series used is as shown in Table 2. Correlation between ΔGWS (output) and a particular precipitation dataset (input) can be considered to be a measure of the persistence of the effect of that input integrated over the timestep. The degree of correlation between ΔGWS and annual ∫PCPA is thus indicative of
the duration of HM in the aquifer system.

| Time series: | Timestep $\tau$: |
|---|---|
| PCP & PCPA | 1 month |
| PCPA (HY) | 1 year |
| ∫PCPA (HY) | 1 year≤$\tau$≤14 years (*upper limit set by length of dataset*) |

**Table 2:** The timestep, $\tau$, for each of the precipitation time series investigated in the study




### 2.4. Regional-Scale Hydrogeology:

In an exploration of climate-groundwater dynamics using GRACE data, the lack of direct physical

observational data means that it is necessary to demonstrate that results are not simply artefacts of

modelling and signal processing (Rodell et al., 2009). The role of hydrogeology in determining

groundwater dynamics is widely acknowledged (Befus et al., 2017; Cuthbert et al., 2019a; de Vries

and Simmers, 2002; Lanen et al., 2013). Here, we seek to validate results inferred from GRACE data

with reference to the physical characteristics of specific aquifer systems. In order to categorise the

hydrogeology of each aquifer system, a number of available global datasets were sourced as raster

files and interrogated in QGIS using the aquifer vector files from WHYMAP LASW. Examined datasets

include:

1.  Groundwater Response Time (GRT) (Cuthbert et al., 2019a)

2.  Hydraulic Conductivity (K) and Porosity ($\Phi$) GHLYMPS high resolution maps (Gleeson et al.,

2014)

3.  Water Table Depth (WTD) (Fan et al., 2013)

As defined above, the GRT is a temporal measure of the latency of aquifer systems that is derived

from their scale and physical properties via Eq. (4). This measures relies on the other datasets listed

for its calculation (Cuthbert et al., 2019a).  K and $\Phi$ are high-resolution datasets derived from

recently developed lithological maps of the Earth's surface (Hartmann and Moosdorf, 2012) and

their computation uses established geological parameters (Gleeson et al., 2014). However, K is

based on permeability mapping from hydrolithologies that have a standard deviation of ~2 orders of

magnitude (Gleeson et al., 2011). WTD is 30 arc-second (~1km.) resolution dataset compiled from

available observational data extended by modelled interpolation (Fan et al., 2013). All of these

datasets are global and derived from combinations of observations and modelled data.


### 3.0. Results:

The main results for each aquifer system are given as a monthly time series of ΔTWS and ΔGWS vs.

PCP and an annual time series of ΔGWS vs. PCPA and ∫PCPA, shown as **Figure 2 a-r** for dryland

systems and **Figure 3 a-j** for humid systems. The outcomes are summarised in **Table 3**. As a general





result, all time series plots show a qualitative relationship between ΔGWS and PCP that exhibits interesting and potentially important spatio-temporal variations. The quantitative results show that for ΔGWS there is a strong correlation with annual ∫PCPA for aquifer systems in dryland environments whereas in humid environments, the strongest correlation is with monthly PCP. Three

aquifers – Guarani Aquifer, Indus River Basin and Canning Basin - do not follow this general classification and anomalies are discussed further in section 4.3 below, and in the SI.

| Aquifer System | Monthly PCP vs ΔTWS | Monthly PCP vs ΔGWS | Monthly PCPA vs ΔGWS | Annual PCPA vs ΔGWS | Monthly ∫PCPA vs ΔGWS | Annual ∫PCPA vs ΔGWS | Aridity Class | Aridity Index | GWS Net Change over SRP |
|---|---|---|---|---|---|---|---|---|---|
| Upper Kalahari | 0.64 (2) | 0.47 (2) | 0.13 (2) | *0.22* (2) | 0.67 (2) | **0.88** (2) | Semi-Arid | 0.42 | Increasing |
| Karoo | 0.15 (7) | 0.25 (7) | *0.07* (7) | *0.21* (7) | 0.71 (7) | **0.88** (7) | Semi-Arid | 0.28 | Increasing |
| Senegal | 0.67 (2) | 0.55 (2) | 0.15 (2) | *0.14* (2) | 0.61 (2) | **0.87** (2) | Semi-Arid | 0.20 | Increasing |
| California Central Valley | 0.53 (2) | 0.46 (2) | 0.26 (2) | 0.56 (2) | 0.60 (2) | **0.84** (2) | Semi-Arid | 0.22 | Decreasing |
| Great Artesian | 0.45 (2) | 0.33 (2) | 0.34 (2) | 0.67 (2) | 0.61 (2) | **0.80** (2) | Arid | 0.18 | Stable |
| North China Plains | 0.34 (2) | 0.22 (2) | 0.18 (2) | *0.26* (2) | 0.65 (2) | **0.80** (2) | Dry Sub-Humid | 0.57 | Decreasing |
| Umm Ruwaba | 0.87 (2) | **0.83** (2) | *0.12* (2) | 0.55 (2) | 0.20 (2) | 0.64 (2) | Semi-Arid | 0.33 | Stable |
| Congo | 0.67 (2) | **0.67** (2) | *0.11* (3) | 0.43 (3) | 0.27 (3) | 0.62 (3) | Humid | 1.22 | Stable |
| Maranhao | 0.82 (2) | **0.75** (2) | 0.30 (2) | 0.74 (2) | *0.11* (2) | 0.40 (2) | Humid | 0.91 | Decreasing |
| Indus River | 0.30 (1) | 0.11 (1) | 0.19 (3) | **0.37** (3) | 0.15 (3) | *0.34* (3) | Arid | 0.16 | Decreasing |
| Amazon | 0.88 (2) | **0.82** (2) | *0.08* (2) | -0.12 (2) | 0.13 (2) | *0.33* (2) | Humid | 1.99 | Stable |
| Guarani | 0.50 (3) | 0.48 (3) | 0.42 (3) | **0.78** (3) | *0.01* (3) | *0.26* (3) | Humid | 0.90 | Increasing |
| Ganges-Brahmaputra | 0.75 (2) | **0.69** (2) | 0.06 (2) | 0.03 (2) | 0.03 (2) | *0.01* (2) | Humid | 0.86 | Decreasing |
| Canning | 0.35 (2) | 0.19 (2) | 0.15 (3) | *0.26* (3) | -0.15 (3) | *-0.01* (3) | Arid | 0.13 | Decreasing |
| Indus River post '08 | 0.42 (1) | 0.15 (1) | 0.21 (3) | 0.73 (3) | 0.34 (3) | 0.89 (3) | Arid | 0.16 | Decreasing |
| Canning post '06 | 0.41 (2) | 0.24 (2) | 0.22 (3) | 0.61 (3) | *-0.02* (3) | *0.24* (3) | Arid | 0.13 | Decreasing |

**Table 3: Summary Table of Results from Monthly & Annual Time Series & Aridity Datasets.**
Summary of all correlation results from time series datasets [Pearson Correlation Coefficient & (lag in months)] and the aridity indices derived from the CGIAR-CSI Global-Aridity dataset (Trabucco and Zomer, 2019). ΔGWS trend over SRP also shown. Results in italics fall below the t-test threshold. Aquifers are ranked in order of Pearson Correlation Coefficient for Annual ∫PCPA vs ΔGWS. For each

Aquifer system the strongest ΔGWS correlation with PCP or PCPA is shown in bold. Truncated time series results shown for 2 systems.









**Figure 2**: Monthly ΔTWS & ΔGWS vs PCP and Annual ΔGWS vs ∫PCPA Time Series for each of the dryland climate zone aquifer systems, as labelled. Systems are ordered by decreasing PCC for annual ΔGWS vs ∫PCPA. All time series are plotted to the aquifer system lag as set out in Table 3, where ΔTWS (ΔGWS) lags PCP (PCPA) by the specified number of months. Y-axis units are Equivalent Water Height (EWH) in cm. Note the variation in the y-axis scales. 7 of the annual ∫PCPA data series have been scaled x10 for clarity, where indicated.


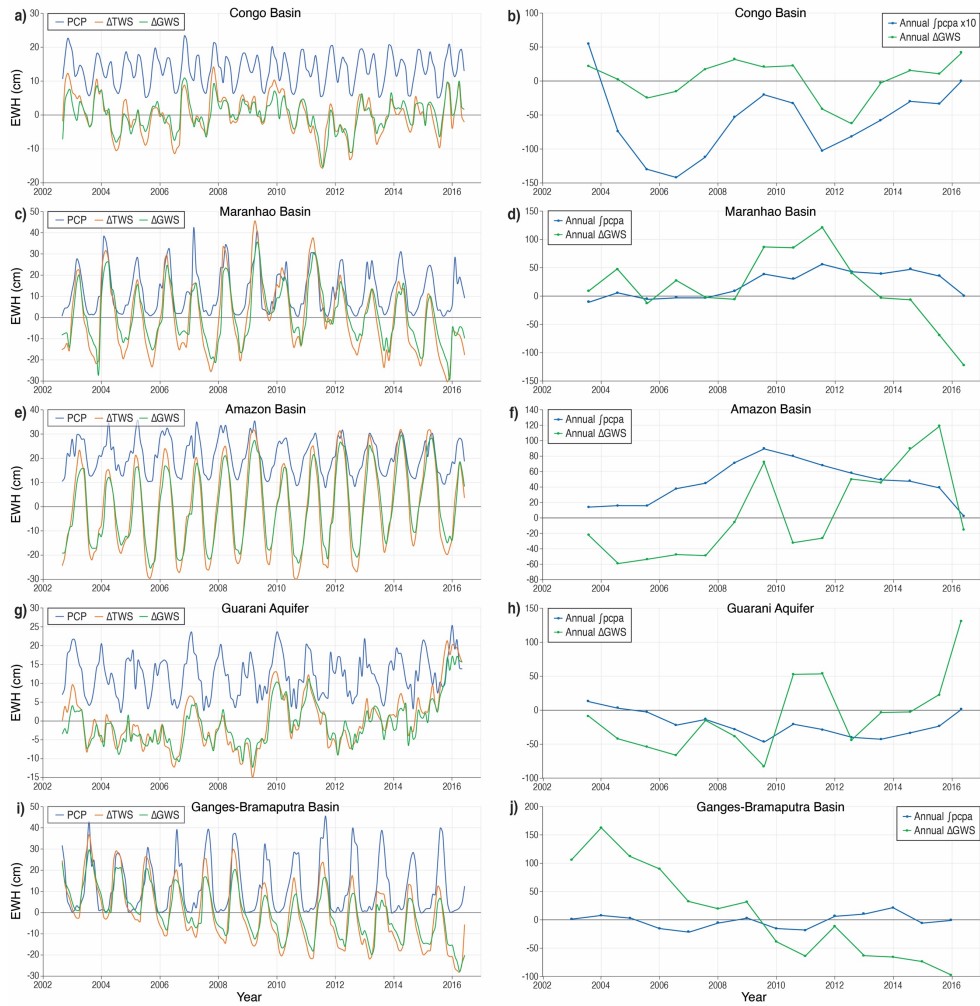

**Figure 3**: Monthly ΔTWS & ΔGWS vs PCP and Annual ΔGWS vs ∫PCPA Time Series for each of the humid climate zone aquifer systems, as labelled. Systems are ordered by decreasing PCC for annual ΔGWS vs ∫PCPA. All time series are plotted to the aquifer system lag as set out in Table 3, where ΔTWS lags PCP by the specified number of months. Y-axis units are Equivalent Water Height (EWH) in cm. Note the variation in the y-axis scales. Congo Basin annual ∫PCPA data series has been scaled x10 for clarity.





GRT, shown in **Figure 4**, is a measure of the time it takes for an aquifer system to equilibrate after a
        change in boundary conditions, as discussed above. For the 14 studied aquifers, it extends from
        centennial to millennial timescales as indicated from median values reported in **Tables 4 and S1**. For
        humid aquifers, GRT ranges from 100 to 350 years whereas for dryland systems GRT then escalates
        to values well in excess of 1,000 years for semi-arid and arid basins; the sub-humid North China

Plains Aquifer has a GRT of ~550 years. This order of magnitude point of transition can be identified
        as the threshold between sensitive (rapid) and insensitive (slow) aquifer response times (Cuthbert et
        al., 2019a), which show a broad global relationship with aridity. This observation helps to explain
        groundwater storage responses to climate variability through the memory of the aquifer system
        defined by both physical characteristics and geographical location. The role of HM is discussed

further in section 4.1.

| Aquifer System | Aridity Classification | Aridity Index | Annual ∫PCPA vs ∆GWS [PCC] (lag in months) | GRT: log (GRT) (GRT in yrs) |
|---|---|---|---|---|
| Indus River post '08 | Arid | 0.16 | 0.89 (3) | 3.96 |
| Upper Kalahari | Semi-Arid | 0.42 | 0.88 (2) | 2.95 |
| Karoo | Semi-Arid | 0.28 | 0.88 (7) | 5.74 |
| Senegal | Semi-Arid | 0.20 | 0.87 (2) | 5.70 |
| California Central Valley | Semi-Arid | 0.22 | 0.84 (2) | 3.01 |
| Great Artesian | Arid | 0.18 | 0.80 (2) | 6.33 |
| North China Plains | Dry Sub-Humid | 0.57 | 0.80 (2) | 2.74 |
| Umm Ruwaba | Semi-Arid | 0.33 | 0.64 (2) | 4.42 |
| Congo | Humid | 1.22 | 0.62 (3) | 2.12 |
| Maranhao | Humid | 0.91 | 0.40 (2) | 2.55 |
| Indus River | Arid | 0.16 | *0.34* (3) | 3.96 |
| Amazon | Humid | 1.99 | *0.33* (2) | 2.03 |
| Guarani | Humid | 0.90 | *0.26* (3) | 2.20 |
| Ganges-Brahmaputra | Humid | 0.86 | *0.01* (2) | 2.10 |
| Canning | Arid | 0.13 | *-0.01* (3) | 6.46 |

**Table 4:** Relationship between Aridity Index, Climate and Regional-Scale Hydrogeology: Data linking

climate and regional-scale hydrogeology to GW dynamics. *(Italicised results fall below t-test threshold.)*



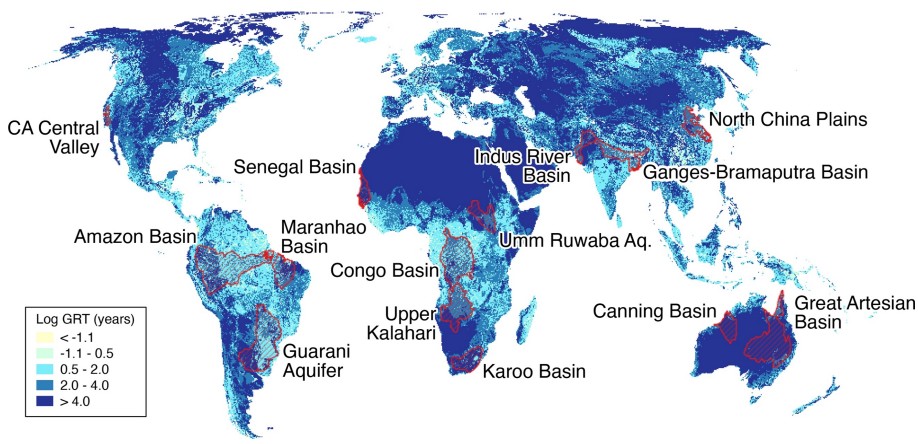


**Figure 4: 14 of the World's Large-scale Aquifers** (the Study Aquifers) overlaid on the GRT dataset [*original dataset from: (Cuthbert et al., 2019a)*]

Presented results represent the outcome of a detailed analysis of the available datasets and, as such, contain important assumptions that need to be acknowledged here. Firstly, the allocation of lag time has been done on a 'best-fit to the ΔTWS data' basis. It is therefore not derived from analysis of intrinsic physical characteristics of the aquifer systems but is consistent with the range of theoretical values derived from hydrodynamic first principles that anticipate a maximum lag time of

3 months for systems with a large GRT (Townley, 1995), as has been observed by Ahmed et al. (2011). Time lags have been tested for consistency through alignment of specific events in the various time series (Storch and Zwiers, 2001). The evident anomaly of a 7-month lag time for the Karoo Basin is discussed in the SI. Secondly, the restricted duration of the GRACE dataset should be acknowledged, particularly with regard to the annual time series. In mitigation, statistical

significance appears to be robust when tested (Zwiers and von Storch, 1995) and the use of PCPA and ∫PCPA datasets is designed to minimise the effect of seasonal climate and short-term trends in ΔGWS (Craddock, 1965). Thirdly, the use of Eq. (3) to derive ΔGWS from GRACE ΔTWS data represents a temporal and spatial approximation in representing sub-surface hydrological processes. Simply put, all water below the soil zone neither necessarily comprises GWS nor will it all eventually

reach GWS due to lateral flow processes. However, on the scale of the aquifer systems considered here, the use of Eq. (3) is a reasonable approximation (de Vries and Simmers, 2002).




### 4.0. Discussion:

#### 4.1. Role of Hydraulic Memory (HM):

A key finding of this study is that GRACE-derived ΔGWS correlates most strongly with annual ∫PCPA

for large-scale aquifers in dryland environments of the tropics and sub-tropics whereas GRACE-

derived ΔGWS correlates most strongly with monthly PCP in humid environments at these latitudes.

Further, we show that there is correspondence between the annual ∫PCPA vs ΔGWS correlations and

GRTs of large-scale aquifer systems (**Table 4**); the latter is a measure derived in accordance with Eq.

(4) (Cuthbert et al., 2019a).  HM ultimately derives from the physical properties of the saturated

portion of the aquifer system (Townley, 1995) and system memory as measured by Eq. (5) is

representative of the physical properties of an aquifer system and its climate. Long and Mahler

(2013), for example, used a total of 16 metrics to describe particular North American Karst aquifer

systems in their IRF ($\theta$ in Eq. (5)). In contrast, von Asmuth and Knotters (2004) used 4 parameters to

describe groundwater dynamics in their transfer function that they argue represents a more

accurate description of the physical system than previously used parametric methods. Further, their

description of groundwater dynamics is capable of accommodating non-stationary elements such as

climate change and groundwater abstraction (von Asmuth and Knotters, 2004). HM as measured by

Eq. (5) is therefore representative of both spatial and temporal variability in aquifer systems but HM

itself can vary spatio-temporally. Indeed the response time to a given boundary change can vary

according to the physical circumstances, with persistence lasting from months to hundreds of

thousands of years (Cuthbert et al., 2019a).

        In this study, the GRACE dataset is not long enough to allow detailed IRF modelling of

aquifer systems based on ΔGWS data, which would require an observational record longer than the

system memory (Long and Mahler, 2013). An extended GRACE series together with reduced

uncertainty in the permeability dataset from which GRT is derived, may generate closer numerical

matches between GRT (Eq. (4)) and HM as measured by the method of this study (Eq. (5)).

Nevertheless, we show that aquifer responses to anomalous precipitation, discussed below, exhibit

long HM in dryland environments and relatively short HM in humid environments. The

correspondence with GRT extends the classification to two broad categories: dryland

environment/long HM/slow GRT and humid environment/short HM/rapid GRT. Note that these

categories represent a simplification of the correspondence between HM derived from the study

datasets and GRT, which in fact exhibits a spectrum in which Umm Ruwaba (dryland), Congo Basin





and Maranhao (both humid) occupy an intermediate position in terms of the correlation between

ΔGWS and annual ∫PCPA, as can be seen from **Table 3**. Aquifers in humid environments, with

exception of the Congo Basin, generally exhibit less HM in this study than expected from GRT values.

These humid aquifers, as can be seen from **Figure 4**, have some of their area with GRTs in the order

of years to tens of years, perhaps meaning that a disproportionate amount of groundwater

processes may be moving through these lower GRT areas. This may explain why humid regions have

less HM overall than is implied by their median GRT.

### 4.2 Aquifer Responses to Anomalous Precipitation:

The annual time series of ΔGWS vs ∫PCPA for each aquifer have been examined to identify years in

which the maximum annual increase in ΔGWS occurred, as identified by the steepest positive

gradient of the ΔGWS line (**Table 5**). These years of extreme recharge, inferred from the increase in

ΔGWS, are then further categorised by whether: (1) prior to the event ∫PCPA is negative, indicating

anomalously dry conditions when soil moisture deficits (SMD) are likely to be widespread; and (2)

the ∫PCPA is concurrently shifting from a negative to positive cumulative anomaly, associated with

an extreme rainfall event. Finally, the NINO3.4 index for 2002-2016 (Huang et al., 2015) has been

examined (KNMI Climate Explorer, 2018) to indicate the state of ENSO, the dominant control on

equatorial precipitation, at the time of the recharge. Nearly all recharge events in dryland aquifer

systems take place at a time of negative ∫PCPA (likely SMD), with most coinciding with extreme

rainfall as recently observed in a pan-African study by Cuthbert et al., (2019b). Extreme recharge

events also generally coincide with El Niño/La Niña events indicating an association with large-scale

modes of climate variability identified previously in tropical Africa (Kolusu et al., 2019; Taylor et al.,

2013b). In contrast, extreme recharge in humid aquifer systems is consistently associated with

neither negative ∫PCPA (likely SMD), nor anomalous rainfall, though the latter is correlated with

ENSO state.







| Aquifer Systems grouped by AI: Dry | Year of Extreme Recharge | Negative ∫PCPA (likely SMD) [Y/N] | ∫PCPA Phase Change [Y/N] | ENSO State |
|---|---|---|---|---|
| Senegal | 2010 | Y | Y | La Nina |
| Umm Ruwaba | 2014 | Y | Y | Neutral |
| U. Kalahari | 2008/9 | Y | N | La Nina |
| Karoo | 2010/11 | N | N | La Nina |
| California CV | 2015/16 | Y | Y | El Nino |
| Indus River | 2003 | Y | Y | El Nino |
| Indus River | 2015 | Y | Y | El Nino |
| Great Artesian | 2010/11 | Y | Y | La Nina |
| Canning | 2010/11 | Y | Y | La Nina |
| North China Plains | 2003 | Y | Y | El Nino |
| Aquifer Systems by AI: Humid | | | | |
| Ganges | 2003 | N | N | El Nino |
| Ganges | 2011 | Y | Y | Neutral |
| Amazon | 2008/9 | N | N | La Nina |
| Amazon | 2011/12 | N | N | La Nina |
| Maranhao | 2008/9 | N | N | La Nina |
| Guarani | 2009/10 | Y | N | El Nino |
| Guarani | 2015/16 | Y | Y | El Nino |
| Congo | 2012/13 | Y | N | Neutral |

**Table 5:** Aquifer systems grouped by AI – Dry (Upper) and Humid (Lower). Extreme recharge years identified from annual time series by slope of ΔGWS plotted line. SMD status inferred by prior negative ∫PCPA and annual ∫PCPA phase change also derived from the same time series. ENSO state
from NINO3.4 Index (Huang et al., 2015).

*4.3 Anomalous Trends in Groundwater Storage:*

Over the SRP determined by the availability of GRACE data, six aquifer systems show a net decline in groundwater storage: California Central Valley, North China Plains, Maranhao, Ganges, Indus & Canning Basins. Of these, two aquifer systems (Indus River and Canning Basins) do not show a strong correlation between ΔGWS and any of the precipitation data series. **Table 4** shows that these same two aquifers do not fit the general classification of the 14 aquifer systems into either




dryland/slow GRT/long HM or humid/rapid GRT/short HM systems. These anomalous characteristics

may reflect  groundwater storage decline through the escalation of groundwater abstraction

referenced previously (Wada et al., 2014) and this hypothesis was tested through further analysis as

follows below and in further detail in the SI.

        The Indus River and Canning Basins superficially present similar stories of groundwater

storage decline yet contextual analysis of their respective GRACE/CLM ΔGWS datasets reveals two

quite different realities. The Indus River Basin supports a population of ~210 million people

(Immerzeel et al., 2010) and its hydrology is strongly influenced by water supply from upstream of

the basin, much of it intended for irrigation (Immerzeel et al., 2010). Surface water is augmented by

groundwater abstraction, which supplies ~31% of the total irrigation demand, but it has been

estimated that ~84% of the groundwater abstracted returns to the aquifer system as leakage from

canals and intensively irrigated fields (Cheema et al., 2014). A net calculation of these effects on

ΔGWS, which is detailed in the SI, shows that the underlying climate-groundwater dynamics are

consistent with the GRT derived from the regional-scale hydrogeology of the aquifer system. In

contrast, the Canning Basin is sparsely populated and is not a centre of agriculture (Richey et al.,

2015). It is, however,  a source of freshwater for iron-ore extraction in adjacent areas (Western

Australia Department of Water, 2011) and very little of the abstracted groundwater is returned to

the aquifer system as its use in mining causes it to become contaminated (Western Australia

Department of Water, 2013). This contaminated groundwater is subsequently disposed in the sea or

evaporation ponds (Prosser et al., 2011). The Canning Basin has a very slow GRT and, situated in an

arid environment, is subject to low rates of groundwater recharge so that the physically sustainable

rate of groundwater abstraction is expected to be very low (Scanlon et al., 2006). The analysis of the

Indus and Canning Basins is evidence of how groundwater depletion, which has been reported

elsewhere (e.g. Famiglietti, 2014; Rodell et al., 2009), impacts relationships between precipitation

and ΔGWS.


***5.0 Conclusions***:

Strong correlations are found between GRACE-derived annual ΔGWS and ∫PCPA for large-scale

aquifer systems in dryland environments. This correlation is much weaker for large-scale aquifer

systems in humid zones where a stronger correlation generally exists between monthly ΔGWS and

monthly PCP. We propose that the correlation between annual ΔGWS and ∫PCPA demonstrates the

existence of hydraulic memory which is central to large-scale climate-groundwater dynamics. For the

studied aquifer systems, the measure of correlation between annual ΔGWS and ∫PCPA also shows



very good correspondence with the groundwater response time, a measure of the hydraulic memory of an aquifer system derived from its regional-scale hydrogeological and catchment properties (Cuthbert et al., 2019a). The 14 aquifer systems can be broadly categorised into two groups, with each group listed in ascending order of groundwater response time:

• Group 1: Dryland/Long HM/slow GRT: North China Plains, Upper Kalahari, California Central Valley, Indus River, Umm Ruwaba, Senegal-Mauritanian, Karoo, Great Artesian & [Canning] Basins

       • Group 2: Humid/Short HM/rapid GRT: Amazon, Ganges, Congo, Guarani, & Maranhao Basins

       Aquifer systems in Group 1 may be less sensitive to seasonal climate variability but also
vulnerable to long-term trends from which they will be slow to recover. In contrast, aquifers in Group 2 may be more sensitive to seasonal climate disturbances such as ENSO-related drought but may also be relatively quick to recover. These characteristics can be applied to anticipate the groundwater response to present conditions and to future pressures that can be expected from anthropogenic climate change (Taylor et al., 2013a). The results from the analysis of GRACE data are
reconciled to regional-scale hydrogeological conditions, which gives confidence in their validity (Beven and Germann, 2013), albeit with the caveat regarding the uncertainties inherent in all the datasets used (Wilks, 2016).

       The new GRACE follow on (GRACE-FO) project has now been launched (Frappart and Ramillien, 2018; Tapley et al., 2019), providing an opportunity to augment the existing GRACE ΔTWS
dataset without recourse to modelling (Ahmed et al., 2019) and to give greater certainty in linking climate-groundwater dynamics to decadal and longer timescale climate systems including the Pacific Decadal Oscillation and Atlantic Multidecadal Oscillation (Wunsch, 1999). An extended dataset will improve the calibration of HM as it relates to specific aquifer systems, providing a robust context for monitoring ΔGWS, including groundwater decline, in real time and protecting fundamentally
important groundwater resources.

*Competing Interests:* The authors declare no competing interests.

*Authors' Contributions:* SO led the analysis of datasets originally compiled by MS and MC, supplemented by datasets developed by SO; RT, CB and MS contributed to the original design of the study with key modifications made by SO; SO drafted the manuscript with input from RT; all authors contributed to, and commented on, revisions to the submitted manuscript.



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

**Data Availability**

Supplementary information is available for this paper as a single PDF file. Data generated and used

in this study can be made available upon request to the corresponding author.
