# Peer review of "Climate-groundwater dynamics inferred from GRACE and the role of"

_Earth System Dynamics, 2019_

## Referee Comment (RC1) · Anonymous Referee #1 · 18 Feb 2020

This is an interesting study that I suspect will be of wide interest to ESD readers. The premise for the research is straightforward and clearly articulated, with a generally well-reasoned account of methods and presentation of findings. I have some relatively minor points to consider for a redrafted version of this manuscript, as follows:

1. There are quite a large number of acronyms used in the manuscript. Not all of these seem essential, and in places they detract from the overall readability. Some are rarely used (e.g. COV, PCC), others are arguably not really necessary (PCP for precipitation). I recommend the authors rethink and reduce the number of acronyms accordingly.

2. Inconsistent capitalisation of nouns. Some terms that are converted to acronyms are not capitalised (e.g. global hydrological models, GHMs – p2, line 52), whereas others are (Groundwater Storage, ∆GWS).

[Figure]

3. P2, line 61-62. Presumably this increase of 15%/decade is a global average?

4. P7, lines 168 and 189. It is potentially somewhat misleading to provide coefficient of variation statistics for sample sizes of 3 and 4. The mean, standard deviation (and thus the coefficient of variation too) are not particularly informative descriptors of central tendency or distribution of samples that are this small.

5. P7, line 192. Improper use of the term "significant" when discussing the magnitude of statistical terms. From my reading of this sentence, "substantial" would be a more appropriate descriptor.

6. Figure 3. The different y-axis scales, and scaling of the data are noted in the caption, but this twin approach to scaling, plus the relatively small size of the panes and faint axis labels makes it somewhat difficult to compare scales of variability between the different aquifers.

7. P16, line 400. The reference for this statement is slightly misleading. It could be read that Zwiers and Von Storch (1995) found robust statistical significance, but actually I think the authors mean that robust statistical significance was found using the methods described by Zwiers and Von Storch.

---

## Author Comment (AC1) · 9 Mar 2020

Referee 1 Comments – Authors' responses:

This is an interesting study that I suspect will be of wide interest to ESD readers. The premise for the research is straightforward and clearly articulated, with a generally well-reasoned account of methods and presentation of findings. I have some relatively minor points to consider for a redrafted version of this manuscript, as follows:

R1 : We appreciate the positive comments regarding the overall manuscript and the constructive suggestions to improve the manuscript. Responses to individual comments are given below.

There are quite a large number of acronyms used in the manuscript. Not all of these

[Figure]

seem essential, and in places they detract from the overall readability. Some are rarely used (e.g. COV, PCC), others are arguably not really necessary (PCP for precipitation). I recommend the authors rethink and reduce the number of acronyms accordingly.

R2 : We will seek to reduce the number of acronyms in the final manuscript including some of the suggested that are rarely used. Use of PCP is expected to remain as we apply this in the context of identifying both PCP and PCPA (anomalies) and consider these to be clear, sensible abbreviations.

Inconsistent capitalisation of nouns. Some terms that are converted to acronyms are not capitalised (e.g. global hydrological models, GHMs – p2, line 52), whereas others are (Groundwater Storage, $\Delta$GWS).

R3 : This comment is helpful in identifying necessary corrections that we will make in the course of the review of acronyms.

P2, line 61-62. Presumably this increase of 15%/decade is a global average?

R4 : We will amend to clarify the text as '...rapid escalation in global groundwater extraction at an average of ~15% per decade...'

P7, lines 168 and 189. It is potentially somewhat misleading to provide coefficient of variation statistics for sample sizes of 3 and 4. The mean, standard deviation (and thus the coefficient of variation too) are not particularly informative descriptors of central tendency or distribution of samples that are this small.

R5 : We acknowledge that the population (not sample) size is small for the use of co-efficient of variation. We nevertheless consider this parameter has merit in explaining divergence among the respective datasets; we propose to include a phrase acknowledging the small population sizes.

P7, line 192. Improper use of the term "significant" when discussing the magnitude of statistical terms. From my reading of this sentence, "substantial" would be a more appropriate descriptor.
R6 : We agree and will amend to read 'substantial' and check all other uses of the word, 'significant' (3 instances), are appropriate.

Figure 3. The different y-axis scales, and scaling of the data are noted in the caption, but this twin approach to scaling, plus the relatively small size of the panes and faint axis labels makes it somewhat difficult to compare scales of variability between the different aquifers.

R7 : We appreciate this comment and will look at ways to improve the clarity of what is being expressed in Figure 3 and improve axis labels. As the primary purpose of the figure is to illustrate correlation between change in GWS and precipitation anomaly for individual systems, we have scaled the data to make these correlations clear. Further information on the variability between aquifers is given in the final two columns of Table 1.

P16, line 400. The reference for this statement is slightly misleading. It could be read that Zwiers and Von Storch (1995) found robust statistical significance, but actually I think the authors mean that robust statistical significance was found using the methods described by Zwiers and Von Storch.

R8 : We agree with that this clarification is necessary and will amend to state: 'when tested using the methods described by Zwiers and Von Storch (ref.)'

---

## Short Comment (SC1) · 2 May 2020

This study presents an interesting analysis that examines the relationship between climate forcing and groundwater response on 17 major aquifers globally. The results are important, indicating strong relationships between groundwater storage and monthly precipitation in humid settings and with annual precipitation in more semiarid regions. These results have important implications for management, indicating the importance of seasonal variations in storage in humid regions and much longer-term variability in semiarid regions and time to recover from extreme events. The findings are consistent with results of the authors previous studies on episodicity of recharge in semiarid regions in Africa. The finding that recharge events in dryland aquifers occur during negative cumulative PCPA is very interesting and coincidence with

extreme rainfall is very important. The linkage to ENSO is also very valuable (L. 460 – 470). I agree with the use of the JPL mascons solution for GRACE data and the use of NOAH land surface model considering the large variability among the GLDAS models. The following includes some minor comments: In the abstract the authors refer to ENOS as seasonal; however, I think of ENSO as more interannual with 3 – 5 yr timescales. The authors refer to baseflow from groundwater sustaining rivers and wetlands being fundamentally important, especially in semiarid and arid regions; however, one should recognize that in many semiarid regions surface water recharges groundwater. L. 45: The authors suggest that future management of freshwater resources will be a critical issue linked to climate change but I think it is already a critical issue because of climate extremes (droughts and floods). L. 63 – 65: In contrast to the statement from Wada et al., 2014; recent reports suggest that water use has been stable or decreasing in past decades in the U.S. and China. Zhou, F., et al. (2020), Deceleration of China's human water use and its key drivers, Proceedings of the National Academy of Sciences, 117(14), 7702-7711. https://www.usgs.gov/special-topic/water-science-school/science/trends-water-use-united-states-1950-2015?qt-science_center_objects=0#qt-science_center_objects L. 90: This is a good point on sources of uncertainty with application of GRACE data to groundwater storage. L. 175 – 180: GRACE uses a baseline from 2004 – 2009; however, I think it would be better to calculate anomalies for your data based on your entire record. It is not clear why you don't combine JPL and CSR mascons? JPL relies on models to process the GRACE data whereas CSR does not. The combination should be more robust. L. 237: It would be good to use a consistent time period for calculating all anomalies throughout the paper. L. 310: I think it is important to indicate the uncertainties in these global datasets. For example, the water table depth map developed by Fan differs markedly from that developed by the British Geological Survey for Africa. L. 423: Long and Mahler (2013) applied the analysis to karst aquifers, which are similar to surface water drainage systems. These systems differ markedly from many aquifer systems.

---

## Author Comment (AC2) · 13 May 2020

Authors' responses to Bridget Scanlon's Interactive Comment:

We thank Bridget Scanlon for her comments and support for the general intent of the paper. We are encouraged that, apart from minor comments, the methods, results and findings of the paper are considered robust and useful. We appreciate that the appended comments are helpful and intended to improve the manuscript.

This response is also provided in the form of a pdf supplement.

We provide the following responses to specific comments:

Abstract:

[Figure]

'In the abstract the authors refer to ENSO as seasonal; however, I think of ENSO as more interannual with 3 – 5 yr timescales'

We agree and propose simply to omit the word 'seasonal'. The original reason for the inclusion of this adjective was the known (inter-annual) periodicity in the control that ENSO exerts on seasonal precipitation (i.e. heavy rains, drought).

L40 – L42:

'The authors refer to baseflow from groundwater sustaining rivers and wetlands being fundamentally important, especially in semiarid and arid regions; however, one should recognize that in many semiarid regions surface water recharges groundwater.'

This characterisation is misrepresentative and a regretted oversight on our part; we propose to amend the wording as follows:

"Baseflow from groundwater sustains rivers and wetlands in the absence of rainfall and is therefore fundamentally important to the ecology of semi-arid and arid regions in particular. . ."

to be replaced by:

"Bidirectional flows between surface water and groundwater are fundamentally important to the ecology of semi-arid and arid regions (drylands), where surface water often recharges groundwater and baseflow from groundwater can sustain rivers and wetlands in the absence of rainfall"

L45:

'The authors suggest that future management of freshwater resources will be a critical issue linked to climate change but I think it is already a critical issue because of climate extremes (droughts and floods)'

Agreed – we propose to amend this sentence:

"Future climate change in which anthropogenic emissions of greenhouse gases transform patterns of natural variability, together with substantial socio-economic change, predicate that management of freshwater resources will become a critical task ((Famiglietti, 2014)."

to read:

"Climate change in which anthropogenic emissions of greenhouse gases transform patterns of natural variability, together with substantial socio-economic change, predicates that management of freshwater resources has and will increasingly become a critical task (Famiglietti, 2014)."

L63 – 65:

'In contrast to the statement from Wada et al., 2014; recent reports suggest that water use has been stable or decreasing in past decades in the U.S. and China. Zhou, F., et al. (2020), Deceleration of China's human water use and its key drivers, Proceedings of the National Academy of Sciences, 117(14), 7702-7711. https://www.usgs.gov/special-topic/water-science-school/science/trends-water-useunited-states-1950-2015?qt-science_center_objects=0#qt-science_center_objects'

We appreciate this insight and helpful suggestion. Zhou et al. (2020) specifically refer to a slowing in the rate of the increase in water use in China (Zhou et al., 2020) whereas the USGS online report clearly highlights a decline in total freshwater withdrawals in the US. To avoid confusion, we have revised the text here to refer simply to the estimate that relates to global groundwater withdrawals so that it is amended to read:

"In the context of an ∼85% increase in global groundwater abstraction from 1979 to 2010 (Wada et al., 2014), an understanding. . .'

L175-180:

'GRACE uses a baseline from 2004 – 2009; however, I think it would be better to calculate anomalies for your data based on your entire record. It is not clear why you

don't combine JPL and CSR mascons? JPL relies on models to process the GRACE data whereas CSR does not. The combination should be more robust.'

We appreciate these comments and, with respect to the suggested baseline, have conducted the analysis as proposed. Our TWS anomalies are calculated based on the entire record which means they have been adjusted to a time mean baseline taken over the whole period of GRACE data. This text on lines 175 to 178 is intended to convey that methodology:

"GRACE $\Delta$TWS is not a time-invariant measure (Wahr et al., 1998) and in the standard datasets all anomalies are given with respect to a baseline which is the mean over the period January 2004 to December 2009 (JPL NASA, 2019). Here, the completed available GRACE $\Delta$TWS time series is examined with respect to climate anomalies in the same timeframe. Consequently, the JPL-Mascon $\Delta$TWS dataset has been rescaled with respect to a time-mean taken over the whole period of GRACE operation (08.2002 – 07.2016), which is the study reference period (SRP) (JPL NASA, 2019)."

To clarify we will amend this passage to:

"GRACE $\Delta$TWS is not a time-invariant measure (Wahr et al., 1998) and in the standard datasets all anomalies are given with respect to a baseline which is the mean over the period January 2004 to December 2009 (JPL NASA, 2019). However, we examine the completed available GRACE $\Delta$TWS time series with respect to climate anomalies over the consistent timeframe of the entire series. Consequently, the employed JPL-Mascon $\Delta$TWS dataset has been rescaled with respect to a time-mean taken over the whole period of GRACE operation (08.2002 – 07.2016), which is the study reference period (SRP) (JPL NASA, 2019)."

On the use of CSR Mascons, we did consider this and we decided that for this particular use of the JPL Mascons, the strong correlation (rank correlation coefficients mostly >0.9) between the CSR Mascons and JPL Mascons $\Delta$TWS time series, identified by Scanlon et al. (2016) (Scanlon et al., 2016), was sufficient to provide a robust result

using JPL Mascons alone. Specifically, we relied on the following arguments:

1. Since the data are adjusted with respect to the time mean baseline for the entire series, we are not concerned with long-term trends which are the major source of divergence between the JPL and CSR datasets.

2. With the exception of California Central Valley, nine of the analysed aquifer systems are large (>500 km2) and 4 are medium scale (>100 km2), meaning that the correlation between JPL and CSR Mascon datasets is robust due to the scale of the aquifers studied.

3. Our study is concerned with Tropical and Sub-Tropical latitudes where the resolution of JPL Mascons (∼300 km) is equivalent to that of the raw GRACE data, minimising concerns of processing and again improving correlation between the two Mascon datasets.

4. We consider that the greatest source of uncertainty in the GRACE data for this study is derived from the use of one or more LSM's to derive ΔGWS from ΔTWS, as discussed in the paper (L90-93, L188-190)

L237:

'It would be good to use a consistent time period for calculating all anomalies throughout the paper.'

We agree and this is indeed what we have done in the paper. To make this point as clear as possible, we will amend the current text:

"PCPA = monthly precipitation anomalies with respect to the time-mean baseline for the study reference period (SRP – 2002-16)"

to read:

"PCPA = monthly precipitation anomalies with the respect to the consistently applied study reference period time-mean baseline, 2002 – 2016."

L310

'I think it is important to indicate the uncertainties in these global datasets. For example, the water table depth map developed by Fan differs markedly from that developed by the British Geological Survey for Africa'

We agree that it is important to indicate the uncertainties in these global datasets and propose to amend the text as set out below. The divergence between the Fan and BGS datasets for Africa is primarily one of scaling in which Fan shows 5 divisions between 0 m and 25 m depth whereas the BGS shows only 2 divisions (Fan et al., 2013; MacDonald et al., 2012). Within the constraints of the BGS scaling, the two datasets apparently agree reasonably well for most of the continent. For this study, the relevant exception is the Karoo Basin where the two datasets disagree; BGS data show some shallower areas of water-table depth that do not appear in the Fan dataset. We recognise that the response of the Karoo Basin to recharge is to some extent anomalous in this study but we do not explore the reasons in any detail.

Proposed amendment:

L318 on: 'However, K is based on permeability mapping from hydrolithologies that have a standard deviation of ∼2 orders of magnitude (Gleeson et al., 2011) and this variance underlies the uncertainty in each of these datasets used. WTD is 30 arc-second (∼1km) resolution dataset compiled from available observational data extended by modelled interpolation with both these data sources being subject to considerable sampling bias and model uncertainty respectively (Fan et al., 2013). All of these datasets are global and derived from combinations of observations and modelled data.'

L423:

'Long and Mahler (2013) applied the analysis to karst aquifers, which are similar to surface water drainage systems. These systems differ markedly from many aquifer systems.'

The original and current intent of this reference (Long and Mahler, 2013) is to illustrate the use of transfer functions in hydraulic analysis and efficiencies that have been derived in the use of metrics. However, we do not wish there to be any confusion arising from this contextual use of the reference and propose to omit it and to amend L420-426:

"HM ultimately derives from the physical properties of the saturated portion of the aquifer system (Townley, 1995) and system memory as measured by Eq. (5) is representative of the physical properties of an aquifer system and its climate. Long and Mahler (2013), for example, used a total of 16 metrics to describe particular North American Karst aquifer Systems in their IRF ((theta) in Eq. (5)). In contrast, von Asmuth and Knotters (2004) used 4 parameters to describe groundwater dynamics in their transfer function that they argue represents a more accurate description of the physical system than previously used parametric methods."

to read:

"HM ultimately derives from the physical properties of the saturated portion of the aquifer system (Townley, 1995) and system memory as measured by Eq. (5) is representative of the physical properties of an aquifer system and its climate. Von Asmuth and Knotters (2004) use 4 parameters to describe groundwater dynamics in their transfer function ((theta) in Eq. (5)) that they argue represents a more accurate description of the physical system than previously used parametric methods (von Asmuth and Knotters, 2004)."

References:

Famiglietti, J.S., 2014. The global groundwater crisis. Nature Climate Change 4, 945–948. https://doi.org/10.1038/nclimate2425 Fan, Y., Li, H., Miguez-Macho, G., 2013. Global Patterns of Groundwater Table Depth. Science 339, 940–943. https://doi.org/10.1126/science.1229881 Gleeson, T., Smith, L., Moosdorf, N., Hartmann, J., Dürr, H.H., Manning, A.H., Beek, L.P.H. van, Jellinek, A.M., 2011. Mapping

permeability over the surface of the Earth. Geophysical Research Letters 38. https://doi.org/10.1029/2010GL045565 JPL NASA, 2019. Frequently Asked Questions | About [WWW Document]. GRACE Tellus. URL https://grace.jpl.nasa.gov/about/faq (accessed 8.20.19). Long, A.J., Mahler, B.J., 2013. Prediction, time variance, and classification of hydraulic response to recharge in two karst aquifers. Hydrology and Earth System Sciences 17, 281–294. https://doi.org/10.5194/hess-17-281-2013 MacDonald, A.M., Bonsor, H.C., Dochartaigh, B.É.Ó., Taylor, R.G., 2012. Quantitative maps of groundwater resources in Africa. Environ. Res. Lett. 7, 024009. https://doi.org/10.1088/1748-9326/7/2/024009 Scanlon, B.R., Zhang, Z., Save, H., Wiese, D.N., Landerer, F.W., Long, D., Longuevergne, L., Chen, J., 2016. Global evaluation of new GRACE mascon products for hydrologic applications: GLOBAL ANALYSIS OF GRACE MASCON PRODUCTS. Water Resources Research 52, 9412–9429. https://doi.org/10.1002/2016WR019494 Townley, L.R., 1995. The response of aquifers to periodic forcing. Advances in Water Resources 18, 125–146. https://doi.org/10.1016/0309-1708(95)00008-7 von Asmuth, J.R., Knotters, M., 2004. Characterising groundwater dynamics based on a system identification approach. Journal of Hydrology 296, 118–134. https://doi.org/10.1016/j.jhydrol.2004.03.015 Wada, Y., Wisser, D., Bierkens, M.F.P., 2014. Global modeling of withdrawal, allocation and consumptive use of surface water and groundwater resources. Earth System Dynamics 5, 15–15. Wahr, J., Molenaar, M., Bryan, F., 1998. Time variability of the Earth's gravity field: Hydrological and oceanic effects and their possible detection using GRACE. Journal of Geophysical Research: Solid Earth 103, 30205–30229. https://doi.org/10.1029/98JB02844 Zhou, F., Bo, Y., Ciais, P., Dumas, P., Tang, Q., Wang, X., Liu, J., Zheng, C., Polcher, J., Yin, Z., Guimberteau, M., Peng, S., Ottle, C., Zhao, X., Zhao, J., Tan, Q., Chen, L., Shen, H., Yang, H., Piao, S., Wang, H., Wada, Y., 2020. Deceleration of China's human water use and its key drivers. PNAS 117, 7702–7711. https://doi.org/10.1073/pnas.1909902117

Please also note the supplement to this comment:

https://www.earth-syst-dynam-discuss.net/esd-2019-83/esd-2019-83-AC2-supplement.pdf